# Rapid Conversion of Amyloid-Beta 1-40 Oligomers to Mature Fibrils through a Self-Catalytic Bimolecular Process

**DOI:** 10.3390/ijms22126370

**Published:** 2021-06-14

**Authors:** Bertrand Morel, María P. Carrasco-Jiménez, Samuel Jurado, Francisco Conejero-Lara

**Affiliations:** 1Departamento de Química Física, Instituto de Biotecnología e Unidad de Excelencia de Química Aplicada a Biomedicina y Medioambiente (UEQ), Facultad de Ciencias, Universidad de Granada, 18071 Granada, Spain; samuelju@ugr.es; 2Departamento de Bioquímica y Biología Molecular I, Facultad de Ciencias, Universidad de Granada, 18071 Granada, Spain; mpazcj@ugr.es

**Keywords:** oligomerization, aggregation kinetics, fibrillation, mechanism, catalysis, Abeta

## Abstract

The formation of fibrillar aggregates of the amyloid beta peptide (Aβ) in the brain is one of the hallmarks of Alzheimer’s disease (AD). A clear understanding of the different aggregation steps leading to fibrils formation is a keystone in therapeutics discovery. In a recent study, we showed that Aβ40 and Aβ42 form dynamic micellar aggregates above certain critical concentrations, which mediate a fast formation of more stable oligomers, which in the case of Aβ40 are able to evolve towards amyloid fibrils. Here, using different biophysical techniques we investigated the role of different fractions of the Aβ aggregation mixture in the nucleation and fibrillation steps. We show that both processes occur through bimolecular interplay between low molecular weight species (monomer and/or dimer) and larger oligomers. Moreover, we report here a novel self-catalytic mechanism of fibrillation of Aβ40, in which early oligomers generate and deliver low molecular weight amyloid nuclei, which then catalyze the rapid conversion of the oligomers to mature amyloid fibrils. This fibrillation catalytic activity is not present in freshly disaggregated low-molecular weight Aβ40 and is, therefore, a property acquired during the aggregation process. In contrast to Aβ40, we did not observe the same self-catalytic fibrillation in Aβ42 spheroidal oligomers, which could neither be induced to fibrillate by the Aβ40 nuclei. Our results reveal clearly that amyloid fibrillation is a multi-component process, in which dynamic collisions between different interacting species favor the kinetics of amyloid nucleation and growth.

## 1. Introduction

Amyloid aggregation of proteins is behind a variety of diseases of tremendous social and economic impact, including Alzheimer’s disease (AD) [1]. These diseases are characterized by the conversion of peptides or proteins from their soluble functional states to fibrillar aggregates. The presence of extracellular senile plaques and intracellular neurofibrillary tangles are still the main hallmark of AD [2]. These protein deposits, usually known as amyloid, have a highly organized fibrillar structure [3]. The finding that the major constituent of the senile plaques is the amyloid-β (Aβ) peptide, naturally present in the human brain, led researchers to investigate at first Aβ fibril formation, both in vitro and in vivo, as a possible therapeutic target. Aβ is an aggregation-prone polypeptide with 39–43 residues, produced naturally by proteolytic cleavage of the transmembrane amyloid precursor protein (APP).

Although the initial focus of research was set on the prominent amyloid fibrils, multiple reports have demonstrated that the neurotoxic effects could be mainly attributed to soluble and diffuse Aβ oligomers [4]. For instance, even in the absence of mature fibrils, oligomeric species were sufficient to induce neuronal death in mouse models [5,6]. The recognition that early oligomers are the main toxic agents to cells and therefore are likely to contribute very significantly to the onset and spread of disease has promoted a very active investigation of their mechanism of assembly, structural characteristics, and interactions with other biomolecules to identify how they could trigger toxicity [1,7,8,9,10,11,12,13]. Unfortunately, since oligomers consist of a heterogeneous combination of polymorphic intermediates generated by multiple aggregation pathways, it has been very challenging to characterize their structures, properties, and their role in the overall fibrillation process.

The physical basis of amyloid formation has been difficult to describe due to the sensitivity of the protein aggregation processes to experimental conditions (pH, agitation, temperature, concentration, ionic strength, fibril growth, and sample preparation) [14,15,16,17]. However, after decades of extensive efforts, comprehensive models of the kinetic mechanisms of protein fibrillation have emerged [18,19,20,21].

The kinetics of formation of amyloid fibrils by Aβ in vitro, like many other proteins, has been traditionally described using a nucleation-polymerization (NP) mechanism [22,23,24,25], in which rate-limiting formation of an oligomeric nucleus, associated with an aggregation lag phase or nucleation phase, is followed by a rapid growth phase. In the growth phase, monomers are attached to the ends of nuclei or growing fibrils. Fibrillation reaches a plateau when monomers are depleted below the minimal concentration supporting fibril growth. As in all nucleation-dependent processes, the lag phase can be shortened or even removed by addition of fibril “seeds” [22], but also by changes in the environmental conditions or by mutations in the protein sequence that accelerate the nucleation [16,26,27,28,29,30].

However, during the lag phases of fibrillation, a number of low-molecular-weight oligomers with a variety of sizes and morphologies have been described to accumulate [9,31,32,33,34]. These oligomers subsequently form β-sheet–rich oligomeric nuclei and prefibrillar aggregates, which undergo further growth to higher-order aggregates of heterogeneous size and morphology. Despite early oligomers have often been dismissed as minor species or off-pathway aggregates, an increasing number of studies reporting their accumulation has challenged the traditional NP mechanism of Aβ aggregation. Due to their implication in the mechanisms of cytotoxicity, it is important to clearly decipher the initial steps leading to the formation of the amyloid oligomeric precursors.

The nucleated conformational conversion (NCC) mechanism has been more recently proposed as a more general mechanism in the amyloid nucleation route [35,36]. Rapidly forming oligomers undergo a slow conformational transition from largely unstructured aggregates to more organized nuclei, able to subsequently evolve into a β-sheet dominated fibrils [9,35]. However, it has been evidenced for some systems that simple homogeneous nucleation could not account for certain observations in the aggregation kinetics and other nucleation mechanisms, including fibril-catalyzed secondary nucleation and fibril fragmentation, should be also considered [18,37,38]. Despite this increasing knowledge in the kinetics of amyloid fibril formation, the crucial process of conversion from dynamic oligomers to ordered amyloid fibrils still remains unresolved.

We have recently shown that disaggregated Aβ peptides spontaneously form micelle-like oligomers above certain critical concentrations [33]. These micelles strongly accelerate the formation of aggregation nuclei and further develop towards higher order oligomers. These oligomers obtained from Aβ40 could evolve to mature amyloid fibrils depending on their concentrations and incubation time. In contrast, Aβ42 oligomers do not evolve to fibrils so efficiently, but a second type of high-order oligomers becomes favored.

Here, we take advantage of this system able to form rapidly amyloid fibrils in order to further clarify the mechanism of conversion of Aβ40 to fibrillar aggregates. We have investigated the role that different fractions isolated from the Aβ aggregation pool play in the conversion of Aβ oligomers to amyloid fibrils by using different biophysical techniques to monitor the structural and morphological changes and the kinetics of the conversion. Our results reveal a novel mechanism of amyloid fibrillation, in which smallest Aβ species acquire the property of catalyzing a fast and extensive conversion of oligomers to mature amyloid fibrils.

## 2. Results

### 2.1. Effective Nucleation of Amyloid Structure from Disaggregated Aβ Depends on a Dynamic Exchange between Monomer and Oligomeric Species

As described previously, freshly disaggregated LMW-Aβ40 prepared by SEC above a critical micellar concentration (CMC) consists of a mixture of oligomeric micelles and low-order species in dynamic exchange [33]. We demonstrated that the micelles strongly accelerate formation of amyloid nuclei. To explore the role of each of these two components in this process, they were further separated through a 10 kDa-cutoff ultrafiltration unit at 4 °C. The aggregation kinetics from each fraction, corresponding to the retentate (R10-LMW-Aβ40) and the filtrate (F10-LMW-Aβ40) were subsequently followed by ThT fluorescence and compared with the mixture (Figure 1a).

The low molecular weight species (F10-LMW-Aβ40) composed mainly by monomer and dimer of Aβ40 (<10 kDa) do not bind ThT significantly, even after 14 h of incubation at 37 °C. On the other hand, the largest species (>10 kDa) present in the retained fraction (R10-LMW-Aβ40) bind ThT weakly, suggesting the initial presence of some oligomeric β-sheet structures, but upon incubation at 37 °C, we could only observe a small increase in the ThT fluorescence intensity during the first 100 min and then the signal rapidly reaches a plateau. This aggregation kinetics was significantly different from that reported for LMW-Aβ40 at similar concentration, in which the development of ThT fluorescence indicating cross-beta structure takes place rapidly and progressively without lag phase, consistent with a fast formation of oligomeric species containing amyloid nuclei [33]. The signal increase does not reach saturation within the time frame of the experiment.

These results suggest that formation of amyloid nuclei needs a bimolecular interaction between both low molecular weight species (monomer or dimer) and larger aggregates (micelles) in solution to be effective.

### 2.2. Conversion of Aβ40 Oligomers to Fibrils Is Dependent on the Presence of Low Molecular Weight Species

We previously showed that incubation of disaggregated LMW-Aβ40 at 37 °C under micelle favoring conditions produced rapidly spheroidal oligomers that we called type A oligomers (Aβ40-OA) [33]. These oligomers could be purified by SEC and then separated from low-molecular weight species (monomers and dimers) by ultrafiltration through a 10 kDa cutoff, as described in Section 4—Material and Methods. We named the purified oligomers as p-Aβ40-OA and the low molecular weight fraction obtained by ultrafiltration as F10-Aβ40-OA. The p-Aβ40-OA oligomers were able to fibrillate under further incubation at 37 °C, but the extent of fibrillation was however highly dependent on the concentration of the oligomers and the incubation time, suggesting a possible bimolecular character for this fibrillation process.

To investigate the role of the low oligomerization state species, Aβ40-OA oligomers were obtained without the ultrafiltration step, so that the resulting mixture maintains the lowest molecular species. Generally, Aβ40-OA oligomers were obtained by incubation of a sample of disaggregated LMW-Aβ40 at 80 μM during 20 h at 37 °C were then eluted from the SEC column at a concentration around 25 μM. We named this type of oligomer preparations as c-Aβ40-OA (“crude” type A oligomers) to distinguish them from the purified ones (p-Aβ40-OA).

The biophysical properties of the two different oligomer preparations were compared. At the same concentration, freshly prepared c-Aβ40-OA and p-Aβ40-OA had similarly low exposed hydrophobicity, according to Bis-ANS fluorescence intensity (Appendix A). According to DLS measurements, p-Aβ40-OA appeared to have an average initial hydrodynamic radius of 11 nm (Appendix A), whereas c-Aβ40-OA oligomers appeared more expanded and exhibits an average radius of 26 nm (Appendix A). FTIR spectroscopy analysis indicates that c-Aβ40-OA presents a higher proportion of unfolded components and a lower content of β-sheet compared to p-Aβ40-OA (Appendix A and Appendix A). This difference may be attributable to the presence of the soluble, low-molecular weight Aβ40, likely in predominantly unfolded conformation.

The morphology of the oligomers was analyzed by AFM (Figure 2a,b). Freshly obtained c-Aβ40-OA consists of a mixture of spheroidal oligomers of different sizes, similar to those previously reported for purified p-Aβ40-OA [33]. Although the oligomers were similar in shape, most p-Aβ40-OA oligomers had heights between 2 and 5 nm with fewer larger species (6–20 nm height) [33], whereas c-Aβ40-OA oligomers appeared significantly higher with a broader height distribution ranging from 3 nm to 10 nm (Figure 2b). A similar size increase was also suggested by the DLS measurements. This size difference may be caused by centrifugal force applied during the additional ultrafiltration step used to purify the p-Aβ40-OA oligomers. Nevertheless, apart from this size difference, both oligomer preparations had similar spheroidal morphology and only differ in the presence of the low-molecular weight species in c-Aβ40-OA.

Then, we compared the aggregation behavior of the two oligomer preparations upon incubation at 37 °C under identical conditions. The aggregation kinetics monitored by ThT fluorescence clearly shows that at the same concentration c-Aβ40-OA forms amyloid structure much faster and more extensively than p-Aβ40-OA (Figure 1b). As a control, isolated F10-Aβ40-OA did not show any ThT fluorescence increase during incubation at 37 °C indicating therefore no cross-beta aggregation under these experimental conditions. Both p-Aβ40-OA and c-Aβ40-OA increase their exposed hydrophobicity with incubation, but this increase was much higher for c-Aβ40-OA (Appendix A). DLS also shows that c-Aβ40-OA oligomers aggregate much faster (Appendix A) and forms larger aggregates (Appendix A) than p-Aβ40-OA at similar concentration. FT-IR spectra show a strong increase in β-sheet and a significant reduction in unfolded structure within the first 100 min of incubation of c-Aβ40-OA at 37 °C (Appendix A). Although, c-Aβ40-OA acquire less total percentage of β-sheet compared to p-Aβ40-OA (Appendix A), [33], its formation was much faster in agreement with the ThT and DLS experiments.

To better understand the differences between the aggregation mechanism of the two types of oligomers, the aggregation kinetics of c-Aβ40-OA and p-Aβ40-OA was further studied by ThT fluorescence at different concentrations using oligomers preparations obtained by incubation of LMW-Aβ40 for 4 h (Appendix A) or for 20 h (Figure 3), with similar results, consistently with the observation that most of the secondary structure and hydrophobicity changes of LMW-Aβ40 occur within the first 120 min of incubation [33].

The aggregation kinetics was strongly dependent of concentration. We measured the initial rates of aggregation at different concentrations from the slopes of the ThT kinetic traces extrapolated to time zero, as described in previous studies [16,20,21,29] (Figure 3b). Although the range of concentrations explored for c-Aβ40-OA was smaller than that studied for p-Aβ40-OA, the aggregation process was strongly accelerated for c-Aβ40-OA, i.e., oligomers in presence of the species with a molecular weight lower than 10 kDa. Moreover, the apparent orders of the kinetics, derived from the slope of the double logarithmic plots, were 1.8 for p-Aβ40-OA and 2.6 for c-Aβ40-OA. These analyses reveal that the aggregation processes do not follow first-order kinetics and their rates are strongly dependent of the concentration. The higher apparent order observed for c-Aβ40-OA suggests a higher molecularity for the rate-limiting step of the process.

In our previous study, we showed that p-Aβ40-OA forms amyloid fibrils by incubation at 37 °C only at high concentrations (50 μM or more), whereas at intermediate concentrations of around 20–30 μM it only forms protofilaments or nascent fibrils after 1 day incubation at 37 °C [33]. Our DLS, FTIR, and especially ThT experiments indicate much faster aggregation of c-Aβ40-OA towards cross-beta aggregates even at low concentrations. To determine the type of aggregates formed by c-Aβ40-OA, AFM experiments were carried out after a 5-h incubation at 37 °C (Figure 2c–h). Even at relatively low concentration (25 μM), c-Aβ40-OA forms long amyloid fibrils (Figure 2c,e). Their morphology was similar to that of the fibrils formed by p-Aβ40-OA at high concentration, but in the case of c-Aβ40-OA, they appeared to be significantly more abundant and much longer. The fibril height was 6–10 nm and the section profiles show clearly a characteristic twisted morphology with a twist period of around 100 nm (Figure 2f,h), as reported in other studies [39,40]. In addition, a significant amount of spherical oligomers remains in the mixture (Figure 2g).

Taken altogether, these results indicate that fibrillation was strongly enhanced by the low molecular weight species present in the c-Aβ40-OA samples. We cannot exclude from these results, however, that the ultrafiltration process may have physically altered the oligomers and make them less fibrillation competent.

### 2.3. Pure Type A Oligomers (p-Aβ40-OA) Can Behave as c-Aβ40-OA under Certain Conditions

To further clarify the role of the low-molecular weight species in the conversion of Aβ40 spheroidal oligomers to amyloid fibrils, we investigated the ThT aggregation kinetics of p-Aβ40-OA supplemented with the low molecular weight species (F10-Aβ40-OA). The addition of F10-Aβ40-OA at relatively low concentration strongly accelerates the aggregation of p-Aβ40-OA, following comparable kinetics to that obtained for c-Aβ40-OA at similar concentration (Figure 4a). This confirms that the rapid fibrillation capacity of the oligomers can be restored by addition of the low-molecular weight species.

A significant effect on the ThT fluorescence kinetics was also observed when freshly prepared c-Aβ40-OA was enriched with 3.3 μM F10-Aβ40-OA (Figure 4b). The initial growth of ThT fluorescence intensity was faster than freshly prepared c-Aβ40-OA. However, the fluorescence reaches a plateau earlier, at a lower value than expected for a c-Aβ40-OA sample with similar total peptide concentration. These results suggest that addition of the low-molecular weight species increases the amount of growing fibrils but appears to reduce the overall extent of fibrillation.

Our kinetic analysis of the initial aggregation rates shown above indicates a higher order for the fibrillation of c-Aβ40-OA than for p-Aβ40-OA. This difference in apparent kinetic order could be explained by considering that fibrillation of the purified oligomers may need a slow release of the low-molecular weight species, which act then as catalysts of fibrillation. This slow rate-limiting step would reduce the apparent order of the overall fibrillation reaction.

To check this possibility, p-Aβ40-OA was left at room temperature for 40 min after their purification and subsequently the ThT fluorescence aggregation kinetics at 37 °C was measured (Figure 4c). After this pretreatment, p-Aβ40-OA aggregated much faster compared to freshly purified p-Aβ40-OA. Although there were some differences compared to c-Aβ40-OA aggregation kinetics, this experiment indicates that p-Aβ40-OA can generate and release low molecular weight species that then accelerate fibrillation, suggesting therefore an autocatalytic process.

These results demonstrate that type-A oligomers and low-molecular weight species are in dynamic exchange and that both components are necessary for fibrillation.

### 2.4. Low-Molecular Weight Species Act as Catalyst of Fibrillation

Then, we sought to elucidate if purified oligomer samples of p-Aβ40-OA, pre-incubated at 37 °C until reaching a plateau in ThT fluorescence, could undergo additional fibrillation if they were supplemented with different concentrations of F10-Aβ40-OA. Figure 5 reveals that addition of a small amount of F10-Aβ40-OA to p-Aβ40-OA triggers a rapid and strong increase of ThT fluorescence indicating fast fibrillation (Figure 5a,b). This fluorescence increase was accompanied by extensive formation of β-sheet structure, as indicated by CD and FT-IR experiments (Appendix A).

The effect of addition of F10-Aβ40-OA to preincubated c-Aβ40-OA was much less intense (Figure 5c,d), as expected because initial fibrillation had already progressed to a higher plateau since c-Aβ40-OA contains low molecular weight species. The observed effects of F10-Aβ40-OA on oligomer fibrillation suggest two possible mechanisms: (a) a rapid addition of the low molecular weight Aβ to the ends of already existing amyloid fibrils or nuclei formed by the preincubation of Aβ40-OA; or (b) a rapid conversion of preformed Aβ nuclei in spheroidal oligomers to fibrils catalyzed by the F10-Aβ40-OA species. We favor the second interpretation because the amplitude of the ThT fluorescence increase was not dependent of the amount of F10-Aβ40-OA added to the oligomers, but increases with the concentration of pure oligomers (Figure 5a,b). Moreover, the large intensity of the secondary structure changes observed was not compatible with the relative small amounts of added F10-Aβ40-OA.

Then we investigated if F10-Aβ40-OA was effective in triggering fibrillation of disaggregated LMW-Aβ40. Supplementation with F10-Aβ40-OA was performed at various pre-incubation times of LMW-Aβ40 at 37 °C at a concentration above the CMC and the extent of amyloid aggregation was monitored by ThT fluorescence (Figure 6a). These experiments reveal that fibrillation was not triggered on freshly prepared LMW-Aβ40 but only on pre-incubated samples and the extent of fibrillation increases with the time of incubation, in agreement with the notion that amyloid nuclei within Aβ40-OA oligomers are the substrates of this conversion.

To investigate the influence of micelles on this fibrillation mechanism, LMW-Aβ40 was pre-incubated at 37 °C for 400 min at two different concentrations, above and below the CMC (60 μM) [33], and then F10-Aβ40-OA was supplemented to the mixture (Figure 6b). Above the CMC, F10-Aβ40-OA triggers fast and extensive fibrillation, whereas below the CMC, the effect was much less significant, likely because of a low amount of Aβ40-OA oligomers formed at this concentration. Since incubation of LMW-Aβ40 above its CMC at 37 °C during 400 min does not form any significant amount of fibrils but only spheroidal oligomers [33], these results demonstrate that F10-Aβ40-OA species act as a catalyst of fibrillation only on the type A oligomers and not on Aβ micelles or soluble disaggregated Aβ species.

We also investigated whether the capacity to trigger fibrillation already resides in the F10 fraction of disaggregated LMW-Aβ40 or this activity was acquired during the aggregation process. To this aim, we carried out similar supplementation experiments but using the low molecular weight fraction obtained by 10 kDa-cutoff ultrafiltration from freshly prepared LMW-Aβ40 (F10-LMW-Aβ40). This fraction should also contain mainly disaggregated monomeric and dimeric amyloid beta peptides.

F10-LMW-Aβ40 produces only a small effect on the fibrillation of pre-incubated LMW-Aβ40, in sharp contrast with the results obtained with F10-Aβ40-OA (Figure 6c). Supplementation experiments with F10-LMW-Aβ40 were also made with c-Aβ40-OA (Figure 6d). No significant effect on the fibrillation of the oligomers was observed confirming that F10-LMW-Aβ40 does not possess fibrillation catalytic activity.

Taken altogether, these experiments indicate that F10-LMW-Aβ40 and F10-Aβ40-OA had different properties and that the catalytic activity of fibrillation is acquired by the low molecular weight species during the aggregation process. Therefore, the F10-Aβ40-OA species could not be simply considered as composed of monomeric or dimeric disaggregated Aβ40.

### 2.5. Aβ42 Spheroidal Oligomers Cannot Be Catalyzed to Fibrillate

In our previous work, we demonstrated that LMW-Aβ42 could also form a similar type of spheroidal oligomers (Aβ42-OA) but having a lower capacity to fibrillate [33]. Consequently, pre-incubated LMW-Aβ42 at 37 °C for 400 min were supplemented with F10-Aβ42-OA (obtained from p-Aβ42-OA purification) and ThT fluorescence kinetics was followed (Figure 7a). No enhancement of fibrillation was observed by F10-Aβ42-OA addition at any time of pre-incubation. This indicates a lack of catalytic capacity of F10-Aβ42-OA to promote fibrillation.

To further investigate this difference, p-Aβ42-OA was incubated at 37 °C for around 400 min and the sample was supplemented with 2 μM of F10-Aβ40-OA. No effect was observed upon addition of the F10 species from Aβ40-OA (Figure 7b). This experiment indicates a specificity of the catalytic process and that F10 species act specifically on oligomers from Aβ40-OA, which could fibrillate efficiently.

### 2.6. Cytotoxic Effects of Aβ40 Oligomers Are Related to Active Fibrillation

We previously established that freshly-prepared LMW-Aβ40 produced a low but detectable cytotoxic effect on SH-SY5Y cell cultures when assayed at concentrations above the CMC but this toxicity disappears if LMW-Aβ40 was pre-incubated at 37 °C, suggesting the conversion to non-toxic Aβ40-OA species [33]. Here we investigated the cytotoxicity on SH-SY5Y cells of c-Aβ40-OA, p-Aβ40-OA, and F10-Aβ40-OA at moderate concentrations (Figure 8). None of the samples produced independently any decrease in cell viability at the concentrations assayed. However, LMW-Aβ40 preincubated at high concentration (120 μM) for 420 min at 37 °C and then supplemented with 2 μM F10-Aβ40-OA immediately before treatment produced significant toxicity, whereas the same preincubated LMW-Aβ40 sample alone did not produce any significant viability decrease. This suggests that active fibrillation catalyzed by F10-Aβ40-OA on the preincubated Aβ40 sample is a major source of cytotoxicity.

### 2.7. Biophysical Characterization of the Lowest Molecular Weight Species F10-Aβ40-OA and F10-LMW-Aβ40

The striking difference in fibrillation catalytic activity between F10-LMW-Aβ40 and F10-Aβ40-OA, prompted us to carry out a comparative biophysical characterization to attempt clarifying the origin of their different properties. However, the yield of F10-Aβ40-OA and F10-LMW-Aβ40 preparations was very low, and the samples were obtained at low concentration, thus limiting the use of some biophysical techniques. In addition, after ultrafiltration, purified samples may further evolve to larger species before or during measurements. These processes may alter the conformations or aggregation states of the preparations making it difficult to interpret the results. Therefore, we have prioritized experiments that did not require further processing of the F10 samples.

To ascertain whether the F10-Aβ40-OA species may have undergone any chemical modification, we analyzed the samples on a Q-TOF mass spectrometer. The molecular masses were identical (Appendix A), indicating that there were no chemical modifications such as covalent dimerization, proteolysis, or oxidation in the F10-Aβ40-OA species. However, this does not exclude some type of isomerization perhaps coming from β-isomerization of Asp residues.

The UV-Vis spectra of F10-LMW-Aβ40 and F10-Aβ40-OA were significantly different (Appendix A). The spectrum of F10-LMW-Aβ40 was similar to the typical spectrum of a tyrosine side chain. However, the absorption band of F10-Aβ40-OA was broader, suggesting a chemical change in the phenol group of the tyrosine side chain or a change in its environment due to intermolecular interactions in oligomeric species.

Intrinsic fluorescence analysis indicates no difference in the tyrosine fluorescence spectra of F10-Aβ40-OA and F10-LMW-Aβ40 (Appendix A) and the absence of di-tyrosine formation (Appendix A) confirms the absence of covalent cross-links from MS analysis.

The morphology of the freshly prepared F10 samples was also analyzed by AFM in dry mode (Figure 9a). Although some large aggregates appear sporadically in both samples, most particles appear small and mainly spherical. The height distribution in F10-LMW-Aβ40 shows, mainly, particles between 0 and 2 nm (Figure 9b), attributable to monomer and dimer. Slightly larger spherical particles of 2–4 nm in height predominate in the F10-Aβ40-OA solutions, suggesting the presence of less Aβ40 monomer, associated to particles below 1 nm [41], and a slightly more aggregated state, possibly dimers (Figure 9b).

Similar size difference was obtained by DLS analysis, although F10-Aβ40-OA was pre-concentrated up to 25 μM using 3 kDa cutoff ultrafiltration filters before analysis as required by the equipment sensitivity (Figure 9c). The size distribution of F10-LMW-Aβ40 spans a range of hydrodynamic radii between 1 and 2 nm, suggesting that F10-LMW-Aβ40 is mainly composed by monomer and dimer of Aβ40. The particle size of F10-Aβ40-OA was however significantly different and appears to be more homogeneous having an apparent hydrodynamic radius around 3.3 nm indicating an oligomeric state. However, the pre-concentration process may have induced this oligomer.

These results did not allow us to discern whether the acquired fibrillation catalytic activity of F10-Aβ40-OA compared to F10-LMW-Aβ40 was due to a conformational difference or a chemical modification. More profound chemical and conformational analysis will be necessary to unequivocally establish the source of this striking property.

## 3. Discussion

### 3.1. A Novel Mechanism of Aβ40 Fibrillation

In this paper, we describe new mechanistic details of the processes by which Aβ40 forms mature amyloid fibrils. Our data highlight the importance of bimolecular processes in both key events leading to fibrillation, that is, the nucleation of amyloid structure leading to oligomeric productive intermediates, and the conversion of these oligomers to mature fibrils. We and others have shown previously that amyloid nucleation is strongly accelerated by the presence of endogenous micellar oligomers [33,34,42], exogenous micelles [17], or lipids vesicles [43]. Formation of on-pathway oligomers and fibrils is accompanied by a progressive increase in β-sheet structure and nano mechanical stability [44,45,46]. Moreover, nucleation of fibril assembly by short peptides has been shown to take place in solute-rich liquid droplets [47]. However, it is unknown how these nucleation events take place. We show here that separating the lowest molecular weight species (monomer and dimer) from larger micellar species of the Aβ40 pool virtually abolish the generation of amyloid nuclei. This suggests that nucleation occurs by collisional events at the surface of the micelles. We previously observed that maximum aggregation rates of α-synuclein induced by SDS micelles occur at SDS concentrations where monomeric α-synuclein coexists with large SDS-protein oligomeric complexes in rapid exchange, suggesting that both states contribute to productive formation of amyloid nuclei [17]. It is then likely that collisions at the micelle surface provide the energy for conversion of disordered Aβ40 to nuclei. Recent theoretical analysis of primary nucleation of Aβ40 shows that it must be a heterogeneous process, occurring at interfaces and not in solution [48].

This and our previous work [33] demonstrates that micelle-mediated nucleation of Aβ40 leads to stable spheroidal oligomers enriched in amyloid nuclei (p-Aβ40-OA in this study). These oligomers, even purified from the low-molecular weight fraction, can be converted to amyloid fibrils upon further incubation, showing that they contain all ingredients for fibrillation. However, this process is rapid only at high oligomer concentrations. The oligomers preparations that have not been depleted of the low-molecular weight species (c-Aβ40-OA) fibrillate much faster and more extensively. Kinetic analysis shows a higher order for the fibrillation in presence of the small species than in their absence, supporting high molecularity of the fibrillation process. This difference in apparent kinetic order can be explained considering that purified oligomers must slowly release the small species that can act then as catalysts of fibrillation. We have shown that purified oligomers can slowly recover the fibrillation property of crude oligomers, confirming that embedded small species that accelerate fibrillation are in fact released from the oligomers. Moreover, addition of these purified small species to the purified oligomers triggers a rapid fibrillation. Interestingly, the extent of fibrillation depends on the oligomer concentration but not on the amount of low-molecular weight species added, indicating a fibrillation catalytic activity of these small Aβ40 species. Therefore, we demonstrate here that efficient fibrillation, such as nucleation, involves a heterogeneous collisional process occurring between small species and oligomers.

The most striking and novel result of this study is the finding that the small species with fibrillation catalytic activity are different from those present in freshly prepared disaggregated Aβ40. Therefore, during the first phase of aggregation some Aβ40 molecules appear to become modified in such a way that they acquire the capacity to catalyze the rapid conversion of oligomers into fibrils. Most interestingly, the substrate of this catalytic fibrillation is not disordered monomeric Aβ40 because fibrillation is not triggered on freshly disaggregated Aβ40, but it needs to be preincubated to generate fibrillation-active oligomers.

Accordingly, under the conditions of our study, Aβ40 fibrillation is a two-step self-catalytic process (Figure 10), in which a first nucleation step converts disordered Aβ40 to fibrillation-competent spheroidal oligomers enriched in fibrillation nuclei. Some of these nuclei are then released from the oligomers to catalyze fibrillation of the rest of Aβ40 oligomer pool.

The autocatalytic replication of protein fibrils emerged since the description of secondary nucleation process [18,49]. However, in this study we have evidenced that auto-catalytic fibrillation can occur not only on the surface of amyloid fibrils but also on the oligomers surface.

### 3.2. Catalyzed Fibrillation Is a Source of Cytotoxicity of Aβ40

We have shown that a strong Aβ40 fibrillation catalyzed by F10-Aβ40-OA species produced detectable cytotoxicity, whereas the separated components did not show significant effects. On-going fibrillation has been described elsewhere as a major source of Aβ neurotoxicity [50,51]. The finding of low molecular weight Aβ species harboring a persistent fibrillation catalytic activity has important implications because the high diffusibility of these small species may confer them a role in prion-like AD propagation [52]. A recent in vivo study using human APP transgenic mice [53] has revealed that oligomeric Aβ alone fails to induce seeded formation of plaque, thus implying the importance of bioavailability of monomeric Aβ species. Moreover, spreading of Aβ deposits in brain was found independent of the oligomers.

### 3.3. Aβ42 Oligomers Do Not Produce Fibrillation Active Species

Despite Aβ42 being able to form a similar type of spheroidal oligomers (Aβ42-OA), they were found to possess less β-sheet and more unfolded structure, and could not form long amyloid fibrils under incubation at comparable concentrations as those of p-Aβ40-OA [33]. Moreover, low molecular weight species extracted from oligomer preparations of Aβ42 (F10-Aβ42-OA) or Aβ40 (F10-Aβ40-OA) are unable to catalyze fibrillation of p-Aβ42-OA. It appears therefore that spheroidal oligomers of Aβ42 are not enriched in active nuclei and therefore incompetent to fibrillate. A similar catalytic specificity was also reported previously using fibrils cross-seeds [54]. This study, focused on the Aβ42 fibrillation, has reported that there is no cross-catalysis of nucleation between Aβ40 and Aβ42 because preformed Aβ42 fibrils fail to nucleate fibrillation of Aβ40 monomers and Aβ40 fibrils do not nucleate Aβ42 monomers fibrillation [55]. Moreover, mixed fibrils were not observed.

In contrast to Aβ40, incubation of LMW-Aβ42 forms a second type of beta-sheet-enriched oligomers (HMW-Aβ42) that may trap the fibrillation active species into stable oligomers. These HMW-Aβ42 oligomers are, not only, toxic by themselves [33,56], but could also act as reservoirs of fibrillation catalytic species.

### 3.4. Are the Fibrillation Catalytic Species Chemical or Conformational Isomers of Aβ40?

An important question arising from this research is the nature of the modification of Aβ40 conferring fibrillation catalytic activity because understanding the nature of the modification could lead to the design to strategies to inhibit Aβ fibrillation targeting these low-molecular weight species.

We found that this different property does not involve molecular weight alteration and, therefore, any chemical modification would be some type of isomerization. For instance, iso-aspartate formation in Aβ enhances fibrillation [57], and in vivo deposits of Aβ have been shown to be enriched in isomeric and racemized Aβ mainly involving aspartic residues [58]. It is also possible that Aβ40 simply acquires a different conformation, although it is difficult to envision how this conformation could remain stable in monomeric or dimeric species under separation and storage, given the disordered nature of Aβ peptides.

It has been already suggested in some particular systems that there could exist some growth-competent and growth incompetent monomers [59,60]. This is particularly the case of human calcitonin, which forms micellar aggregates that have been shown to be rather kinetically inactive species [59]. Such promiscuity in monomer conformation could help in the inhibition of mature fibrils formation by both limiting the availability of growth-competent monomers and through the formation of slowly reversible growth-incompetent species.

Future efforts are needed to ascertain a more detailed understanding and overcome the difficulties in detecting such monomeric reformatting.

## 4. Materials and Methods

### 4.1. Preparation of Disaggregated Aβ Peptides

Synthetic Aβ40 and Aβ42 were obtained from Genecust (Genecust, Europe, Luxembourg) at a purity > 95%. To prepare soluble disaggregated Aβ samples as starting material for aggregation kinetics, we used a SEC purification protocol, as previously described [61]. Lyophilized Aβ was first dissolved in 6 M Gdn-HCl at a concentration of 3 mg mL^−1^ and incubated at room temperature overnight. The solution was centrifuged for 10 min at 14,000× *g* and the resulting supernatant was loaded onto a Superdex 75 HR 10/30 column (GE Healthcare Life Sciences, Chicago, IL, USA) previously equilibrated in 50 mM sodium phosphate, 0.5 mM EDTA, and 100 mM NaCl, with pH 7.4 at a flow rate of 0.4 mL/min. The peak corresponding to disaggregated soluble Aβ (LMW-Aβ) was collected in a pre-cooled, low-binding Eppendorf tube and directly stored on ice. Peptide concentration was determined by measurement of the absorbance at 280 nm using an extinction coefficient of 1490 M^−1^ cm^−1^ [61]. Using this procedure, LMW-Aβ40 solutions of up to 100 µM could be obtained, whereas the yield in LMW-Aβ42 was considerably lower (about 60 μM). Samples at different concentrations were prepared from these stock solutions by direct dilution with buffer. The Aβ samples prepared by this procedure were used immediately.

### 4.2. Preparation and Purification of Oligomer Samples

Disaggregated LMW-Aβ samples, previously obtained by SEC as described above, were incubated at 37 °C for about 20 h (unless otherwise stated) under quiescent conditions to form oligomers as described previously [33,62]. The incubated solution was subsequently injected into a Superdex 75 HR10/30 column (GE Healthcare) and elution peaks were collected. The main elution peak corresponds to “crude” type A oligomers, subsequently called in this manuscript c-Aβ40-OA. The type A oligomers were further purified to remove Aβ in low oligomerization states (monomer and dimer) and concentrated by centrifugal ultrafiltration using a 10 kDa cutoff membrane Amicon Ultra 4 centrifugal filter (Millipore, Darmstadt, Germany). The retained oligomers were washed twice with pre-chilled buffer before further characterization. These purified oligomers are called p-Aβ40-OA in this manuscript.

### 4.3. Formation of Amyloid Aggregates Determined by ThT Fluorescence

To observe the formation of amyloid structure, thioflavine T (ThT) fluorescence was continuously monitored during Aβ aggregation at 37 °C. To start aggregation, a small aliquot of a concentrated stock solution of ThT was added to a freshly prepared disaggregated or oligomeric Aβ sample, to reach a final dye concentration of 10 μM. The sample was then placed into a fluorescence cuvette, which was previously thermostated at 37 °C. Fluorescence intensity of ThT was monitored at 485 nm using an excitation wavelength of 440 nm. Initial slopes of the kinetics were calculated as described previously [20,21].

### 4.4. Atomic Force Microscopy (AFM)

Non-contact mode AFM imaging was performed using an NX-20 instrument (Park Systems, Suwon, South Korea) fitted with pyramidal-shaped silicon cantilevers with a spring constant of 25–75 N/m and a resonance frequency of 200–400 kHz. Peptide sample was diluted with buffer to a concentration of 0.6 μM and a 12 μL aliquot was deposited on freshly cleaved mica and left for adsorption on the substrate for 10–15 min. It was then rinsed three times with MilliQ water (Millipore, Darmstadt, Germany) to remove salts and loosely bound peptide and further dried before imaging. Images were typically acquired as 256 by 256 pixels at a scan rate of 0.5–0.7 Hz. Subsequently, images were processed and analyzed using XEI software (Park Systems, Suwon, South Korea). Representative images of samples were obtained by scanning at least 3 different locations on at least two different samples of the same nature.

### 4.5. Cell Viability Measurements

Neuroblastoma SH-SY5Y cells (Sigma, St Louis, MO, USA) were maintained at 37 °C in a humidified atmosphere of 95% air and 5% CO_2_ and cultured in a 1:1 mixture of Nutrient mixture F12 Ham and Dubelcco’s modified Eagle’s medium supplemented with 2 mM L-glutamine, 100 U mL^−1^ of penicillin, 100 μg mL^−1^ of streptomycin, and 10% fetal calf serum (FCS). Medium was changed every two days. All the cells used in this study were at a low number of passages (<20). For cell viability experiments, cells were seeded onto 96-well plates (20,000 cells per well) and maintained in medium containing 10% FCS for 24 h. Then, the culture medium was replaced with free FCS medium, and the cells were incubated for 24 h with different Aβ preparations or the corresponding volumes of buffer (10 mM HEPES, 100 mM NaCl, and 0.5 mM EDTA at pH 7.4) as a control. For treatment, the different Aβ preparations were previously diluted with an equal volume of medium. After 24 h at 37 °C, neuronal viability was determined using the WST-1 assay from Roche (Cell Proliferation Reagent WST-1). Results were expressed as the percentage of viable cells relative to untreated cells, arbitrary set to 100%. The data are presented as mean ± S.E.M. from at least three independent experiments. A one-way ANOVA was conducted with post hoc comparisons by Scheffe’s test (SPSS 13.0). *p* < 0.05 was considered to be statistically significant.

### 4.6. Particle Size Determination by Dynamic Light Scattering (DLS)

The particle sizes and scattering intensity of Aβ samples were assessed by DLS measurements performed at 37 °C using a Zetasizer μV DLS instrument (Malvern Instruments, Worchestershire, UK). Zetasizer software (Malvern Instruments, Worchestershire, UK) was used in data collection and processing of the correlation function to finally obtain the particle size distributions. Each sample was typically measured 3 times with 15 runs of 5 s.

## 5. Conclusions

In this work, we highlight the importance of bimolecular processes in the key events of amyloid formation. Both nucleation and fibrillation need interplay between oligomers and low-molecular weight species. We report a novel self-catalytic mechanism of fibrillation of Aβ40, in which spheroidal oligomers generate and deliver low-molecular weight species, which have the capacity to catalyze the rapid conversion of the oligomers to fibrils. This fibrillation catalytic property is not present in freshly prepared low-molecular weight Aβ40 and therefore is acquired during the aggregation process. These results could have important implications in the understanding of the mechanisms of amyloid formation, as well as in the pathogenesis of AD.

## Figures and Tables

**Figure 1 ijms-22-06370-f001:**
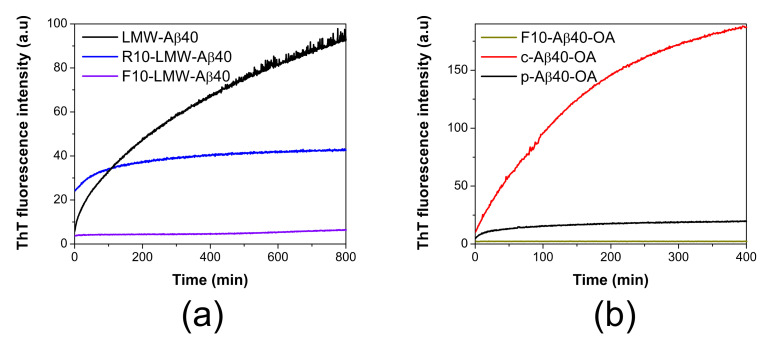
ThT aggregation kinetics of different Aβ40 species. (**a**) Representative aggregation kinetics of LMW-Aβ40 at 90 μM (black line), R10-LMW-Aβ40 at 90 μM (blue line) and F10-LMW-Aβ40 at 50 μM (violet line) followed by ThT fluorescence at 37 °C. (**b**) Comparison of the aggregation kinetics of c-Aβ40-OA (red line) and p-Aβ40-OA (black line) at 24 μM and F10-Aβ40-OA (grey line) at 10 μM followed by ThT fluorescence at 37 °C.

**Figure 2 ijms-22-06370-f002:**
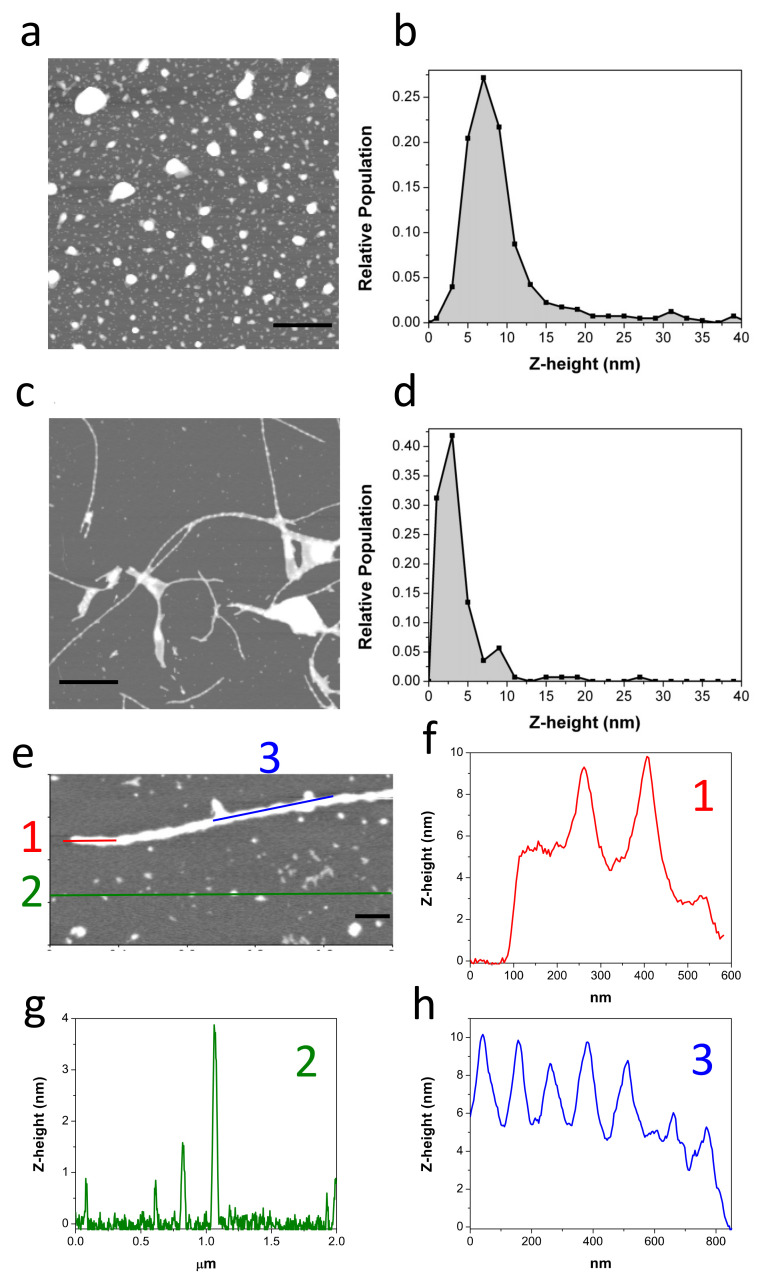
Representative AFM topography image of freshly purified c-Aβ40-OA (c = 25 μM) (**a**). The Z-height (nm) relative frequencies are represented (**b**). Representative AFM topography images of c-Aβ40-OA (c = 25 μM) incubated at 37 °C for 5 h (**c**,**e**). The Z-height (nm) relative frequencies are represented (**d**). Section analysis showing the height distribution along the different colored lines (**f**–**h**). Scale bars represent 1 μm (**a**,**c**) and 200 nm (**e**).

**Figure 3 ijms-22-06370-f003:**
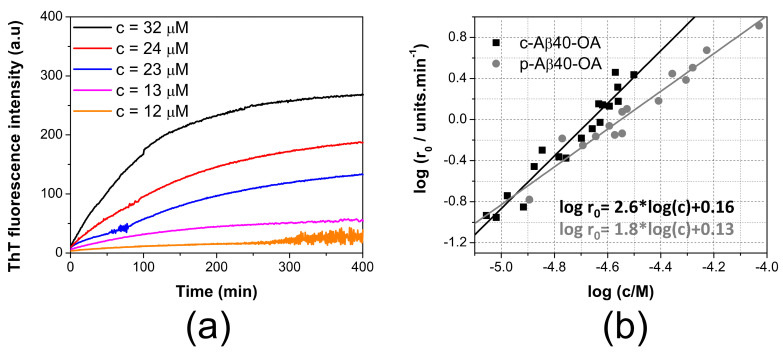
ThT aggregation kinetics followed by ThT fluorescence. (**a**) concentration dependence of the aggregation kinetics of c-Aβ40-OA followed by ThT fluorescence at 37 °C. Crude oligomers were prepared after incubation of LMW-Aβ40 for 20 h. Oligomer concentrations are indicated in the graphs. (**b**) double logarithmic plots of the initial aggregation rates of p-Aβ40-OA (grey symbols) and c-Aβ40-OA (black symbols) determined from the ThT aggregation kinetics versus the initial peptide concentrations. Oligomers samples were prepared individually after incubation of LMW-Aβ40 for 20 h. Symbols correspond to the experimental data. Continuous lines are the best fits to a linear regression.

**Figure 4 ijms-22-06370-f004:**
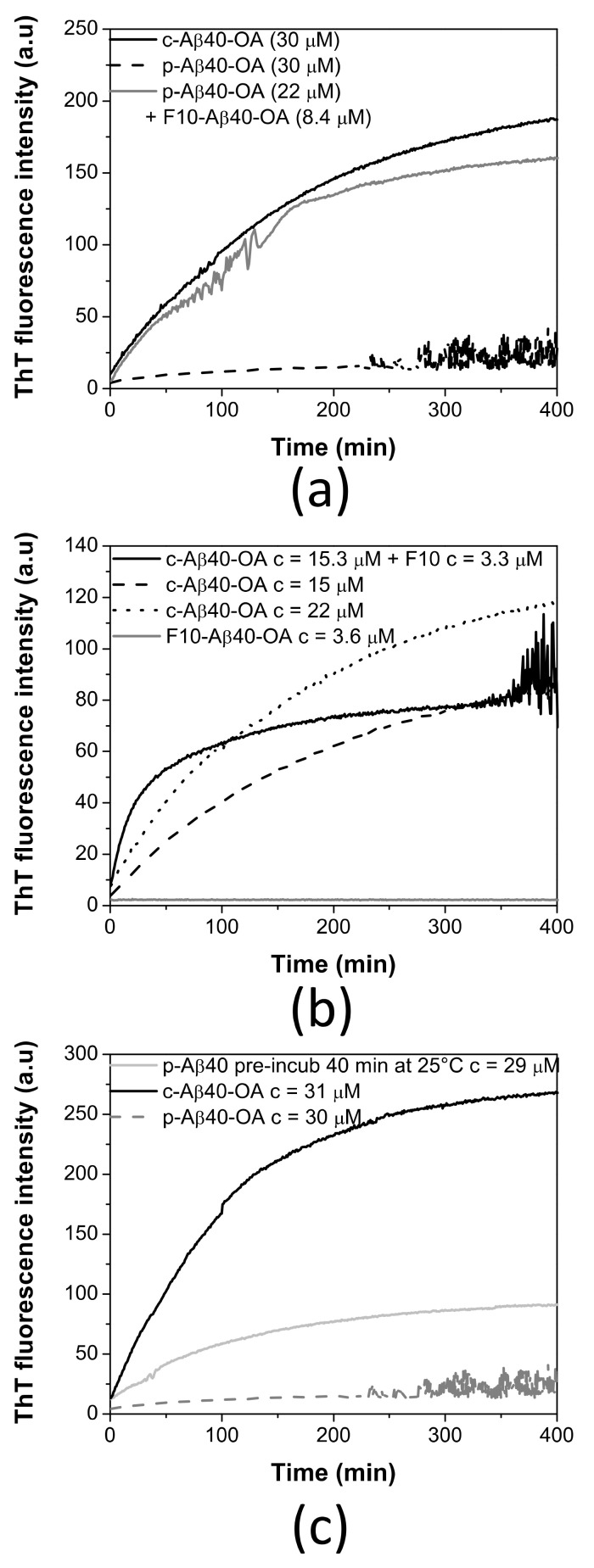
(**a**) Effect of F10-Aβ40-OA addition on freshly prepared p-Aβ40-OA ThT aggregation kinetics. (**b**) Aggregation time dependence of F10-Aβ40-OA at 3.6 μM (grey line), c-Aβ40-OA oligomers at 15 μM (dashed black line), 22 μM (dotted black line), and of c-Aβ40-OA (15 μM) enriched with 3.3 μM F10-Aβ40-OA followed by ThT fluorescence. (**c**) Comparison of the ThT aggregation kinetics at 37 °C of p-Aβ40-OA at 30 μM (dotted grey line), p-Aβ40-OA at 29 μM pre-incubated at 25 °C for 40 min (continuous grey line), and c-Aβ40-OA at 31 μM (black line).

**Figure 5 ijms-22-06370-f005:**
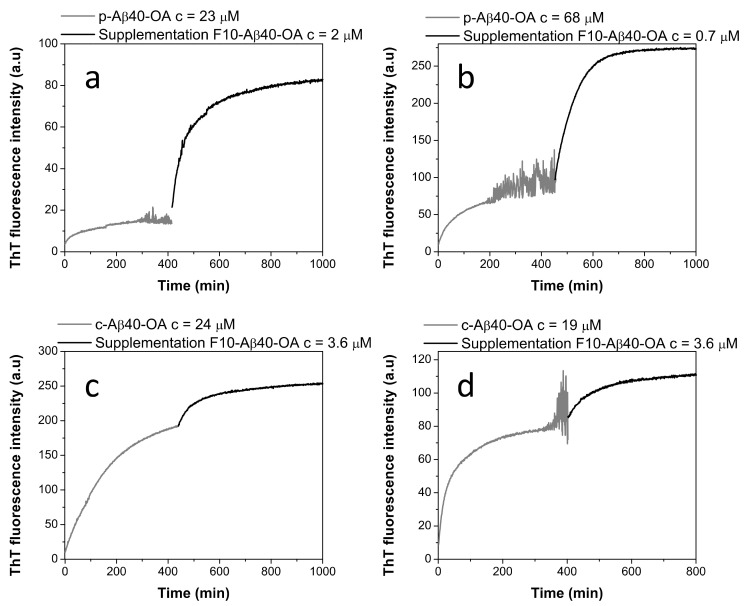
Effect of the supplementation by F10-Aβ40-OA on the aggregation kinetics of p-Aβ40-OA (**a**,**b**) and c-Aβ40-OA (**c**,**d**) followed by ThT fluorescence. The p-Aβ40-OA and c-Aβ40-OA samples were pre-incubated at 37 °C for 400 min and the ThT kinetics were followed prior to supplementation with F10-Aβ40-OA.

**Figure 6 ijms-22-06370-f006:**
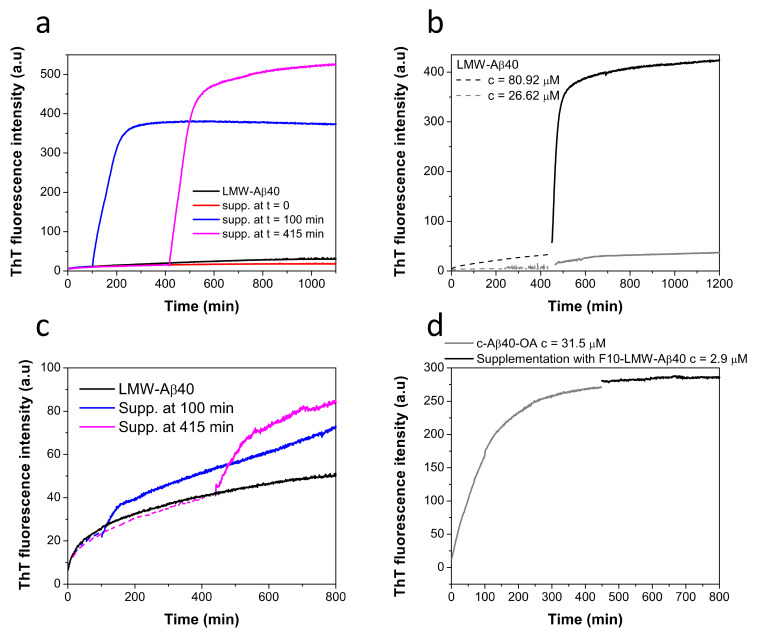
(**a**) Time dependent effect of the supplementation with F10-Aβ40-OA on the aggregation kinetics of LMW-Aβ40 at c = 86 μM (black line) followed by ThT fluorescence. LMW samples were incubated at 37 °C and supplemented with 2 μM of F10-Aβ40-OA at t = 0 (red line), t = 100 min (blue line), and t = 415 min (magenta line). (**b**) Effect of the supplementation with F10-Aβ40-OA on the aggregation kinetics of LMW-Aβ40 at c = 81 μM (black line) and at c = 27 μM (grey line) followed by ThT fluorescence. Samples were incubated at 37 °C for about 400 min (dashed lines) prior to supplementation with 1.9 μM F10-Aβ40-OA (continuous lines). (**c**) Time dependence effect of the supplementation with F10-LMW-Aβ40 on the aggregation kinetics of LMW-Aβ40 at c = 85 μM (black line) followed by ThT fluorescence. LMW samples were incubated at 37 °C and supplemented with 2 μM of F10-LMW-Aβ40 at t = 100 min (blue line), and t = 415 min (magenta line). Data were normalized by sample concentration and experimental points before supplementation are represented in dashed lines. (**d**) Effect of the supplementation with 2.9 μM of F10-LMW-Aβ40 (black line) on the aggregation kinetics of pre-incubated c-Aβ40-OA at c = 32 μM (grey line) followed by ThT fluorescence.

**Figure 7 ijms-22-06370-f007:**
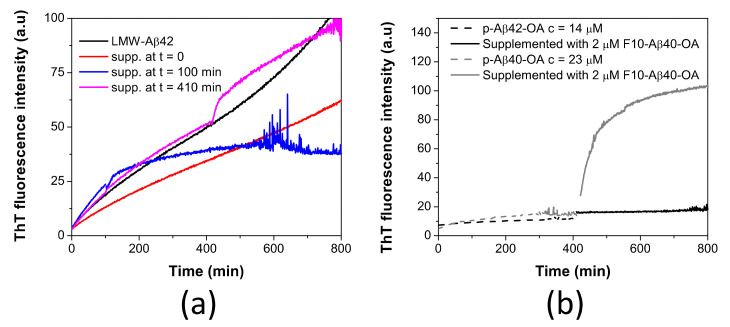
(**a**) Time dependence effect of the supplementation with F10-Aβ42-OA on the aggregation kinetics of LMW-Aβ42 at c = 40 μM (black line) followed by ThT fluorescence. LMW samples were incubated at 37 °C and supplemented with 1.8 μM of F10-Aβ42-OA at t = 0 (red line), t = 100 min (blue line), and t = 410 min (magenta line). (**b**) Comparison of the effect of F10-Aβ40-OA on p-Aβ40-OA (grey lines) and p-Aβ42-OA (black lines).

**Figure 8 ijms-22-06370-f008:**
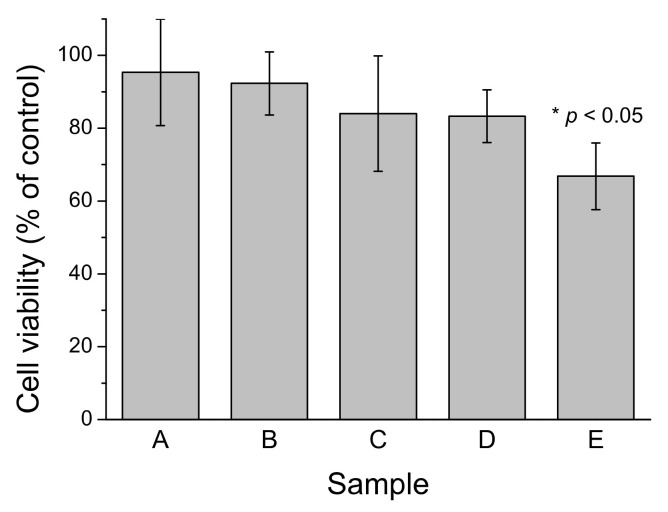
Effect of different Aβ40 preparations on SH-SY5Y cell viability after 24 h treatment. A: c-Aβ40-OA at 20 μM; B: p-Aβ40-OA at 20 μM; C: F10-Aβ40-OA at 12 μM; D: LMW-Aβ40 at 120 μM preincubated 7 h at 37 °C; E: LMW-Aβ40 at 120 μM preincubated 7 h at 37 °C and then supplemented with 2 μM F10-Aβ40-OA. All samples were diluted with an equal volume of culture medium immediately before treatment. Values are mean ± S.E.M. (* *p* < 0.05).

**Figure 9 ijms-22-06370-f009:**
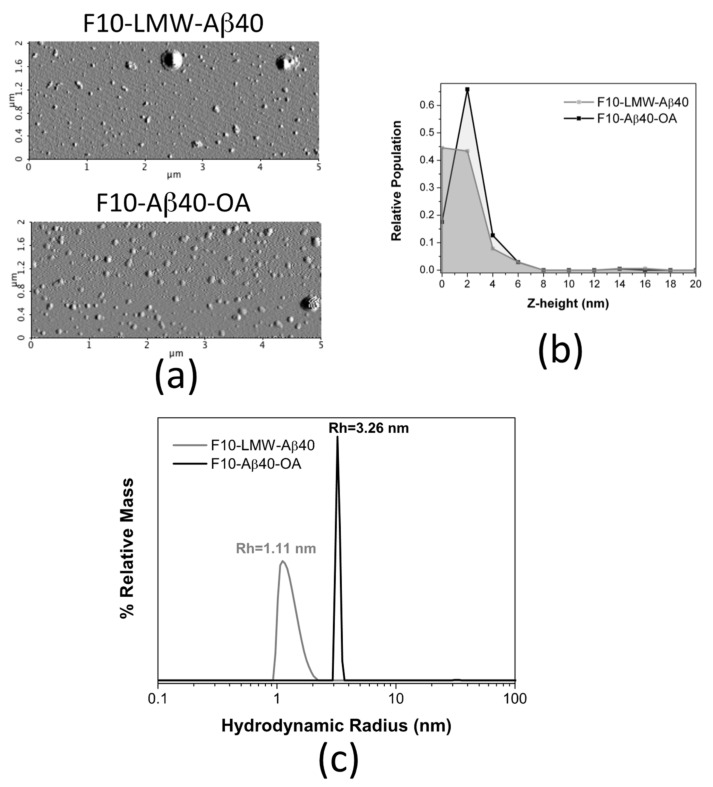
Biophysical properties comparison between F10-Aβ40-OA and F10-LMW-Aβ40. (**a**) Representative AFM amplitude images of freshly prepared F10-LMW-Aβ40 and F10-Aβ40-OA. Relative frequencies of the Z-height (nm) for F10-LMW-Aβ40 (grey) and F10-Aβ40-OA (black) (**b**). (**c**) Particle size distributions obtained by DLS for F10-LMW-Aβ40 (c = 25 μM) (grey line) and F10-Aβ40-OA (c = 25 μM) (black line).

**Figure 10 ijms-22-06370-f010:**
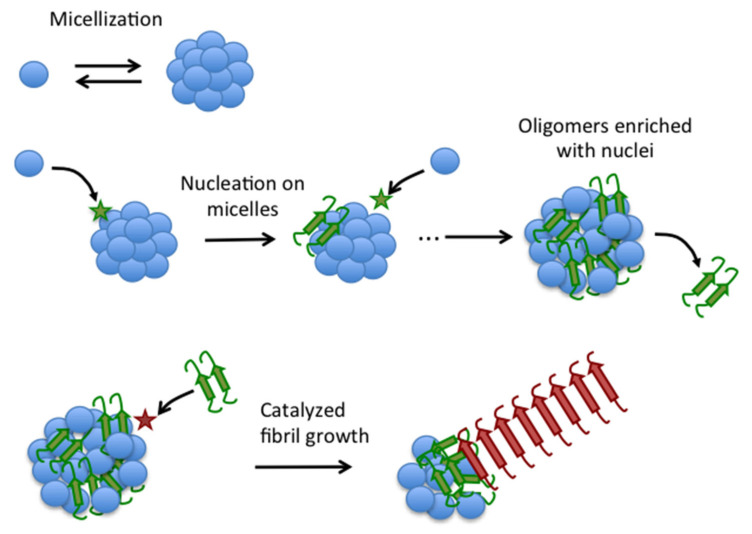
Schematic illustration of the novel autocatalytic mechanism of fibrillation of Aβ40. Blue balls represent disordered Aβ40 monomers in equilibrium with micelles. Aggregation nuclei (green) form by monomer collision with the micelles (star) and remain embedded in the micelles to form type-A oligomers, which can release nuclei to the bulk solution. Nuclei can then catalyze fibrillation on type-A oligomers to form mature fibrils (red).

## Data Availability

Not applicable.

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
