# Peer review of "Rapid Conversion of Amyloid-Beta 1-40 Oligomers to Mature Fibrils through a Self-Catalytic Bimolecular Process"

_ijms, 2021, doi:10.3390/ijms22126370_

Round 1

Reviewer 1 Report

Commmenst for authors are attached as docx file "peer-review-12064205.v1.docx".

Author Response

We would like to thank this referee for his positive feedback.

Reviewer 3 Report

The study by Morel et al. describes the rapid growth of ab40  fibrils  via a self-catalytic process from the oligomer state. This particular observation has not been found in ab42. The authors suggest an interplay between the mechanical stability (via collision) of the amyloid peptides  and the kinetics of the underlying process (nucleation and fibril growth). This observation/hypothesis is novel and deserve to be explored in more detail via molecular simulation or further experiments. I would ask authors to comment in the main text few of my concerns and also improve the quality of the figures as I describe below: 

Comments:

1) The collision hypothesis seems to support their finding, but I would suggest to expand it with more examples from the literature. It is known the role played by the mechanical stability in amyloid oligomers and in fibrils. The typical elastic moduli (a measure of the mechanical stability) is very much meaningful in case of oligomers (~1 GPa) , whereas in amyloid fibrils it ranges between 1-3 GPa. (see for fibril https://doi.org/10.1039/C7CP05269C and https://doi.org/10.3762/bjnano.10.51 and for oligomer https://doi.org/10.1002/ange.201409050). Based on those findings, one can infer an intrinsic nanomechanical stability at the early stage of the aggregation process.    

2) The author should describe more quantitatively parameter of the kinetic processes in text via a table. For instance, the typical rate constant for fibril formation for each species as obtained for each type of ab40/ab42. 

3) Line 106-107: (?) symbols opaque the text.

4) Figure 2, 4, 5 ,6, 7 show symbols (?). Please remove them.

5) Figure 8:  mean value reported for A sample:  ~90±15 which gives some large number above 100. Please clarify in main text.

6) Figure 9: panel (b) please normalise the distributions.

7) Line 531-533: Authors claim the larger beta-content in ab40 monomer over the ab42. In a study by Viet and Li (http://dx.doi.org/10.1063/1.4730410 and references therein) the opposite is shown. A comment is needed.

8) Line 613: authors report the spring constant as ±SD and it should be mean value +/- SD. Please correct there and also for frequency.

9) Figure 10: Authors have to remove again odd symbols and improve the message. Not information in caption about the green and red ab peptides.

10) Finally, the conclusions have to be amended regarding comments above, in particular the nanomechanical evidence in amyloid-beta oligomer and fibrils.

Round 2

Reviewer 3 Report

The author has follow some of my major comments and I am glad about that part of the work. However, the discussion of the mechanical stability in oligomer and fibrils has been narrowed to few lines (34-36) and more over not references are cite.  The authors should take care of the revision more seriously and cite reviews or recent studies (https://doi.org/10.3762/bjnano.10.51 or any other) about the topic.

The images still look clumsy

Author Response

References to mechanical stability were present in the revised form (R1) of the manuscript but we have forgotten to insert them in the "track changes" file of R1.

We have now corrected this mistake, added 3 references (#44, #45, #46 in the main text) and uploaded the R2 versions of the manuscript.

The citations are:

Poma, A. B.; Chwastyk, M.; Cieplak, M., Elastic moduli of biological fibers in a coarse-grained model: crystalline cellulose and Ab amyloids. Physical Chemistry Chemical Physics 2017, 19, (41), 28195-28206, 10.1039/c7cp05269c.

Ruggeri, F. S.; Adamcik, J.; Jeong, J. S.; Lashuel, H. A.; Mezzenga, R.; Dietler, G., Influence of the beta-sheet content on the mechanical properties of aggregates during amyloid fibrillization. Angew Chem Int Ed Engl 2015, 54, (8), 2462-6, 10.1002/anie.201409050.

Poma, A. B.; Guzman, H. V.; Li, M. S.; Theodorakis, P. E., Mechanical and thermodynamic properties of Ab42, Ab40, and a-synuclein fibrils: a coarse-grained method to complement experimental studies. Beilstein Journal of Nanotechnology 2019, 10, 500-513, 10.3762/bjnano.10.51.